# Adaptability and stability analyses of plants using random regression models

**Michel Henriques de Souza**[1]*, **José Domingos Pereira Júnior**[1], **Skarlet De Marco Steckling**[2], **Jussara Mencalha**[1], **Fabíola dos Santos Dias**[1], **João Romero do Amaral Santos de Carvalho Rocha**[2], **Pedro Crescêncio Souza Carneiro**[2], **José Eustáquio de Souza Carneiro**[1]

1 Departamento de Agronomia, Universidade Federal de Viçosa, Viçosa, Minas Gerais, Brazil,
2 Departamento de Biologia Geral, Universidade Federal de Viçosa, Viçosa, Minas Gerais, Brazil

* micheel.1992@gmail.com

**Data Availability Statement:** The dataset are available in the Figshare online repository, at the link: https://doi.org/10.6084/m9.figshare.12668390.v1.

## Abstract

The evaluation of cultivars using multi-environment trials (MET) is an important step in plant breeding programs. One of the objectives of these evaluations is to understand the genotype by environment interaction (GEI). A method of determining the effect of GEI on the performance of cultivars is based on studies of adaptability and stability. Initial studies were based on linear regression; however, these methodologies have limitations, mainly in trials with genetic or statistical unbalanced, heterogeneity of residual variances, and genetic covariance. An alternative would be the use of random regression models (RRM), in which the behavior of the genotypes is characterized as a reaction norm using longitudinal data or repeated measurements and information regarding a covariance function. The objective of this work was the application of RRM in the study of the behavior of common bean cultivars using a MET, based on Legendre polynomials and genotype-ideotype distances. We used a set of 13 trials, which were classified as unfavorable or favorable environments. The results revealed that RRM enables the prediction of the genotypic values of cultivars in environments where they were not evaluated with high accuracy values, thereby circumventing the unbalanced of the experiments. From these values, it was possible to measure the genotypic adaptability according to ideotypes, according to their reaction norms. In addition, the stability of the cultivars can be interpreted as variation in the behavior of the ideotype. The use of ideotypes based on real data allowed a better comparison of the performance of cultivars across environments. The use of RRM in plant breeding is a good alternative to understand the behavior of cultivars in a MET, especially when we want to quantify the adaptability and stability of genotypes.

## Introduction

In the final stages of a breeding program, the most promising lines are evaluated in trials conducted in different environments, composed of different years, places, and seasons. This set of trials, known as multi-environment trials (MET), can provide useful information about the performance of genotypes in environments where we want to recommend a cultivar [1]. In

**Funding:** This work was financed with support resources received by professors Pedro Crescêncio Carneiro and José Eustáquio de Souza Carneiro, these resources being transferred to the Programa Feijão. In addition, the other authors of this work are graduate students who receive resources from three financial support agencies: CNPq (Conselho Nacional de Desenvolvimento Científico e Tecnológico), FAPEMIG (Fundação de Apoio à Pesquisa do Estado de Minas Gerais) and CAPES (Coordenação de Aperfeiçoamento de Pessoal de Nível Superior). There are still research projects funded by these same agencies that assist in conducting experiments in the Feijão Program. Authors: M.H. Souza: scholarship CNPq J. D. P. Júnior: scholarship FAPEMIG S.D.M. Steckling: scholarship CNPq J. Mencalha: scholarship CNPq F.S. Dias: scholarship CNPq J.R.A.S.C. Rocha: scholarship CNPq P.C. Carneiro: research grant CNPq J.E.S. Carneiro: research grant CNPq.

**Competing interests:** NO authors have competing interests.

Brazil, these tests are called Value for Cultivation and Use (Valor de Cultivo e Uso–VCU), and their results are the basis for the recommendation of a new cultivar [2]. In addition, the data from a MET also allows determination of the effect of the genotype by environment interaction (GEI) on the performance of genotypes and make predictions regarding the breeding values of the genotypes in other environments [3]. The evaluation of superior materials across several locations is an essential practice to ensure that the next cultivars have known performance [4].

A method to determine the behavior of genotypes, as well as the effect of the GEI acting on them, is through studies of adaptability and stability. The adaptability is defined as the ability of a genotype to respond advantageously to its environment, while its stability is related to the predictability of its behavior [5, 6]. It is possible to identify genotypes that have wide or specific adaptability to favorable or unfavorable environments. Finlay and Wilkinson [5] defined favorable and unfavorable environments as those that result in the average performance of the genotype being above or below the average of all the trials, respectively.

In recent decades, several methods to evaluate the adaptability and stability have been proposed, based on different statistical principles, such as those methodologies that are based on linear regression models [5–8]. Some of the previous methods to determine the adaptability and stability also included the ideotype concept [9, 10], and resulted in an improved understanding of the relative behavior of the genotypes from a smaller number of parameters. In a review, Eeuwijk et al. [11] show that there are other methodologies to assess the behavior of genotypes that are of note, such as AMMI (Additive Main effects and Multiplicative Interaction) [12] and GGE biplot (Genotype main effects and Genotype x Environment interaction effects) [13].

However, the adaptability and stability analyses still have limitations, especially when used with trials with genetic or statistical unbalanced, heterogeneity of residual variances, and genetic covariance. Another relevant factor is that traditional methodologies for the analyses of adaptability and stability, the behavior of the genotypes is predicted by a linear model adjusted according to an environmental gradient [5, 14], that is, composed of a set of straight lines (one for each genotype in the environments) modeling the G x E interaction in a single dimension [15]. Thus, the predictability of the behavior is compromised if the behavior of the genotypes in the face of environmental variations differs from that predicted by the linear model. Consequently, recommendations based on these methodologies can be biased. An alternative would be the use of Random Regression Models (RRM), as they allow for improved modeling of the behavior of the genotypes. RRM were first proposed by Kirkpatrick et al. [16], and later extended by Schaeffer and Dekkers [17] and Meyer and Hill [18].

The RRM is used mainly in animal breeding, where the phenotypic behavior of animals is characterized by longitudinal data or repeated measurements and information regarding a covariance function [19–22]. The first methods used were based on parametric functions that adjusted regression equations for fixed and random effects [23, 24]. However, these functions have convergence difficulty, mainly in the evaluation of bovine lactation curves, resulting in the search for new functions, with emphasis on orthogonal polynomials. The covariates based on orthogonal polynomials reduce problems with rounding and provide relatively small correlations between the estimated regression coefficients [21]. Among the orthogonal polynomials, Legendre's polynomials, which describe the structures of variation and covariance between genetic and environmental components, are distinct [16, 25] and present computational advantages such as reduced correlations between the estimated coefficients and better convergence properties [26].

In plant science, the application of RRM is recent, as in the works of Sun et al. [27], Ly et al. [28], Momen et al. [29] and Baba et al. [30], where it is used to model the behavior of

individuals along a MET, especially when these trials are on a continuous and gradual scale, in order to capture and predict the variation in the behavior of the genotypes due to environmental changes, even in places where the genotypes have not been evaluated. Thus, these models can be used to select genotypes with responses to environmental variations, maintaining the high ability to predict unmeasured values [27–30]. When considering phenotypic variations in time, as in the case of lactation curves in cattle, the term random regression is common; however, in the case of spatial variation from phenotypic records, as in a MET, the term reaction norm seems to be more acceptable. This term, first described in the field of ecology to describe the natural adaptation of individuals [31], refers to an individual's phenotypic plasticity in response to environmental variation, i.e., a specific relationship between genotype, phenotype, and environmental gradient. The distinction between the terms is subtle, although it is also related to the nature of the variation in the measures [32, 33].

According to Streit et al. [34], one way to treat genotype variation in environments is to consider the multitrait approach, analyzing the information for each environment as a distinct variable. The use of reaction norm models are appropriate to evaluate gradual and continuous variations in the environments, with few parameters and without the need to group individuals in the environments [34]. Thus, knowledge of the reaction norms modeled using Legendre's polynomials can better quantify the adaptability and stability of a set of genotypes evaluated in different environments, aiming for greater accuracy in cultivar recommendations. Therefore, the objective of this investigation was the application of the random regression models in the study of genotypes behavior along a MET, based on Legendre polynomials and genotype-ideotype distances.

## Material and methods

### Genetic material

We evaluated 105 common bean cultivars (*Phaseolus vulgaris* L.), 56 of which were Carioca grains and 49 were Black grains. These cultivars have been recommended in Brazil by breeding programs since 1959. The cultivars used, as well as the institutions of origin and year of recommendation, are listed in S1 and S2 Tables (supporting information).

### Trials

The trials were conducted in different environments (seasons, years, and places), during the dry and winter seasons, between 2013 and 2018, at the Experimental Stations in Coimbra county–Minas Gerais (Unidade de Ensino, Pesquisa e Extensão—UEPE Coimbra: latitude 20˚ 49´44″ S, longitude 42˚45´56″ W and altitude of 713 meters) and Viçosa–Minas Gerais (Aeroporto, latitude 20˚44´38″ S, longitude 42˚50'40" W and altitude of 654 meters; Horta Nova: latitude 20˚45´47″ S, longitude 42˚49´25″ W and altitude of 664 meters; Vale da Agronomia: latitude 20˚46´04″ S, longitude 42˚52´11″ W and altitude of 662 meters), totaling 13 trials. Over the years in which the trials were carried out, the cultivars that were recently launched by the breeding programs were included, thus causing a genetic unbalanced (variation in the number of cultivars in the trials). The 13 trials and their characteristics are listed in S3 Table (supporting information).

The trials were designed in randomized blocks with three replications. The plots consisted of four lines of two meters (m), spaced 0.5 m apart. The treatments used were in accordance with the recommendations for common bean cultures [30]. The evaluated characteristic was grain yield, and they were harvested from the two central lines of each plot. The data were corrected to 13% moisture and converted to kg ha$^{-1}$.

## Statistical analyses

We use RRM to evaluate the behavior of the genotypes as a function that describes this behavior over the gradual and continuous changes in the trials. The genotype's behavior is quantified as their reaction norm. For this, initially, the 13 trials in which the genotypes were evaluated were classified in an environmental gradient, according to the index proposed by Finlay and Wilkinson [5]. According to these authors, the genotype's ability to respond to continuous environmental improvements is a description of its adaptability. The environmental index was determined as follows:

$$I_j = (\overline{Y}_j - \overline{Y}) \tag{1}$$

where $\overline{Y}_j$ is the average of the genotypes $j$-$th$ trial ($j = 1, 2,\ldots, na$, where $na$ is the total number of trials) and $\overline{Y}$ is the general mean. Negative and positive index values indicate unfavorable and favorable trials, respectively. The values of the environmental index were later standardized to the range of orthogonal functions (−1 to 1) to avoid problems of collinearity among the data [35], according to the equation adapted by Schaeffer [36]:

$$I_{js} = -1 + 2\left(\frac{I_j - I_1}{I_{13} - I_1}\right) \tag{2}$$

where $I_{js}$ is the standardized environmental index, $I_j$ is the value obtained in Eq 1 for each trial, and $I_1$ and $I_{13}$ are the index values (Eq 1) obtained for the trials of lowest and highest averages, respectively.

To adjust the behavior of the genotypes along the environmental gradient, it is necessary to test different models of reaction norms. The number of models to be tested depends on the number of trials used (determines the maximum order of the polynomial), the number of effects included in the model via the Legendre polynomials, and the residual covariance structures. Thus, we fitted 14 reaction norm models, where seven were considered with homogeneous residual variance (H) and the other seven with heterogeneous diagonal residual variance (D). The models were fitted with Legendre's polynomials, considering the various polynomial orders, based on the general model, was as follows:

$$y_{ijk} = A_j + R/A_{jk} + \sum_{m=0}^{M-1}\alpha_{im}\Phi_{ijm} + e_{ijk} \tag{3}$$

where: $y_{ijk}$ is the observation of the $i$-$th$ genotype ($i = 1, 2,\ldots, ng$, where $ng$ is the total number of genotypes), in the $j$-$th$ trial ($j = 1, 2,\ldots, na$, where $na$ is the total number of trials), in the $k$-$th$ block ($k = 1, 2, 3$); $A_j$ is the effect of the trial; $R/A_{jk}$ is the fixed effect of the blocks within each trial; $\alpha_{im}$ is the reaction norm coefficient for the Legendre polynomial of order $m$ for the genotypic effects of the genotypes; $\Phi_{ijm}$ is Legendre's $m$-$th$ polynomial for the $j$-$th$ trial, standardized from -1 to +1 for the $i$-$th$ genotype; $M$ is the order of polynomials of the Legendre polynomial for genotypic effects; and $e_{ijk}$ is the residual random effect associated with $y_{ijk}$.

In a matrix, the model above is described as: $y = Xb+Zg+e$, where: $y$ is the vector of phenotypic data; $b$ is the vector of the fixed effects of the combination of blocks × trials added to the general average; $g$ is the vector of genetic effects (assumed to be random); and $e$ is the residual vector (random). $X$ and $Z$ represent the incidence matrix for these effects, respectively. It is assumed that: $g \sim N(0, Kg \otimes I_{ng})$, and $e \sim N(0, \Sigma \otimes I_{np})$, where $I_{ng}$ and $I_{np}$ are identity matrices of the order $ng$ ($ng$ is the total number of genotypes) and $np$ ($np$ is the number of genotypes x the number of blocks), respectively. The symbol $\otimes$ denotes the Kronecker product. $Kg$ is the matrix of covariance coefficients for genotypic effect. $\Sigma$ represents the matrix of residual variances.

After obtaining several models, we choose the one best fit (with lowest mean square error and greater parsimony). For that, some criteria were used, namely: Akaike Information Criterion (AIC) [37], Bayesian Information Criterion (BIC) [38], and Penalizing Adaptively the Likelihood (PAL) [39]. These criteria are described as follows:

$$AIC = -2lnL + 2p \tag{4}$$

$$BIC = -2lnL + pln[n - r(x)] \tag{5}$$

$$PAL = -2lnL + nln(\tilde{n})\frac{\ln(r_n + 1)}{\ln(\rho_n + 1)} \tag{6}$$

where;

$$r_n = 2lnL_{n-1} - 2lnL_1$$

$$\rho_n = 2lnL_{\tilde{n}} - 2lnL_{n-1}$$

and $lnL$ is the logarithm of the likelihood function; $p$ is the number of estimated parameters; $n$ is the number of observations; $r(x)$ is the rank of the fixed effects matrix; and $\tilde{n}$ is the highest number of parameters for the models.

From the chosen model, we utilized the Likelihood Ratio Test (LRT) [40] to test the genetic effects. The LRT is used which is as follows:

$$LRT = -2*(LogL_{mod.r} - LogL_{mod.c}) \tag{7}$$

where: $LogL_{mod.r}$ is the logarithm value of the maximum likelihood function obtained for the reduced model (without the genotypic effect), and $LogL_{mod.c}$ is the logarithm value of the maximum likelihood function obtained for the complete model.

We use the equation proposed by Kirkpatrick et al. [16] to predict the genotypic values ($\hat{g}_{ij}$) for the all genotypes in the trials. The equation is described as:

$$\hat{g}_{ij} = \sum_{m=0}^{M-1}\hat{\alpha}_{im}\Phi_{ijm} \tag{8}$$

where: $\hat{\alpha}_{im}$ is the reaction norm coefficient of order $m$ for the genetic effects of the $i$-th genotype.

Another point that we find relevant is to provide an estimate of the prediction accuracy of the genotypic values, in order to know the reliability of the results. For that, the prediction accuracy was estimated according to the following equation, adapted from Gilmour et al. [41], and according to Kirkpatrick et al. [16].

$$r_{\hat{g}g_{ij}} = \sqrt{1 - \frac{\Phi_{ijm}PEV_{ij}\Phi'_{ijm}}{\Phi_{ijm}\hat{K}_g\Phi'_{ijm}}} \tag{9}$$

where: $r_{\hat{g}g_{ij}}$ is the correlation between the predicted and real genotype values for genotype $i$ in trial $j$, that is, the estimated accuracy; $PEV_{ij}$ is the Predicted Error Variance, obtained from the diagonal elements of the matrix of the estimated coefficients for genotype $i$ in trial $j$; and $\hat{K}_g$ is the covariance matrix of the coefficients, estimated for the genotypic effect.

Finally, after choosing the appropriate reaction norm model and predicting the genotype values, we quantify the individual reaction norm for each genotype aiming to know their adaptability and stability. For that, we used the genotype-ideotype distance (converted into

probability), according to three ideotypes: i) genotypes of general adaptability (genotypes of maximum performance in both unfavorable and favorable environments); ii) genotypes of maximum adaptability to unfavorable environments (genotypes of maximum performance in unfavorable environments, regardless of their performance in favorable environments); and iii) genotypes of maximum adaptability to favorable environments (genotypes of maximum performance in favorable environments, regardless of their performance in unfavorable environments). Each ideotype was based on the phenotypic values of each environment. From the genotypic values, thus, we obtained the value of the genotype-ideotype distance (converted into probability), according to the estimator adapted from Rocha et al. [42], as described:

$$P_{ik} = \frac{\frac{1}{GID_{ik}}}{\sum_{i=1}^{ng} \frac{1}{GID_{ik}}} \qquad (10)$$

where $P_{ik}$ are the probabilities referring to genotype $i$ with regard to ideotype $k$ ($k = 1, 2, 3$; where 1 = genotypes of general adaptability; 2 = genotypes of maximum adaptability to unfavorable environments; and 3 = genotypes of maximum adaptability to favorable environments); and $ng$ is the total number of genotypes. $GID_{ik}$ is the standardized average Euclidean distance for genotype $i$ in ideotype $k$, as given by:

$$GID_{ik} = \sqrt{\frac{\sum_{j}[\hat{g}_{ij} - ide(\hat{g}_{ij})]^2}{nj}} \qquad (11)$$

where, if $k = 1, j = 1,\ldots, na$; if $k = 2, j = 1,\ldots, nd$; if $k = 3, j = 1,\ldots, nf$; $na$ is the highest assumed value for $j$; $nd$ and $nf$ represent the number of unfavorable and favorable environments, respectively; $ide(\hat{g}_{ij})$ is the ideotype drawn from the standardized genotypic values.

It is important to emphasize that the estimators used above also considered the stability of the genotypes' behavior in relation to the ideotype, where the stability can be highlighted as variation regarding the behavior of the ideotype.

We evaluated the performance only in those genotypes that present an accuracy value of at least 80% in the trials, since the accuracy is indicative of the precision in the prediction of genotypic values. Thus, the average accuracy of the trials considered in the cultivar recommendation will also show values equal to or greater than 80%. The standard value is based on that of Resende and Duarte [43], who claimed to have at least 80% accuracy values in cultivar comparison trials.

After obtaining the values of probability of the cultivars, we selected the top ten cultivars (highest probability value) to plot their curves with their respective reaction norms, to view the results. The genotypic value of each cultivar was added, plus the environment average, and the general average, as well as two checks, Pérola (Carioca bean) and Ouro Negro (Black bean), for comparison purposes. These two cultivars were selected as check, as they are used as references for the productivity and quality of grain in consumer markets for the Carioca and Black beans, respectively [44]. The accuracy values (S4 Table) and the values of the genotype-ideotype distance (recommendation probability values—S5 Table) are available in the supporting information.

## Software used

The joint analyses was carried out using ASREML software [45]. The study of the adaptability and stability of cultivars was carried out using R [46] according to the ASReml-R and R Stats packages. The code for the analyses is available in the S1 Code (supporting information).

**Table 1. Trials evaluated with their environmental index.**

| Trial | Description | Environmental index | Standardized environmental index |
|---|---|---|---|
| 12 | Dry/2017/Aeroporto | -1028.89 | -1.00 |
| 9 | Dry/2016/UEPE Coimbra | -868.67 | -0.89 |
| 4 | Winter/2013/Vale da Agronomia | -607.25 | -0.71 |
| 10 | Winter/2016/UEPE Coimbra | -500.76 | -0.64 |
| 8 | Dry/2016/Aeroporto | -466.43 | -0.61 |
| 6 | Winter/2015/UEPE Coimbra | -167.35 | -0.41 |
| 5 | Dry/2015/UEPE Coimbra | 31.02 | -0.27 |
| 2 | Dry/2013/Vale da Agronomia | 106.80 | -0.22 |
| 3 | Winter/2013/Coimbra | 127.71 | -0.20 |
| 7 | Dry/2016/UEPE Coimbra | 259.39 | -0.11 |
| 1 | Dry/2013/Coimbra | 486.60 | 0.05 |
| 11 | Winter/2016/Horta Nova | 758.98 | 0.23 |
| 13 | Winter/2017/UEPE Coimbra | 1868.83 | 1.00 |

## Results

The environmental index values, according to Finlay and Wilkinson [5], are shown in Table 1. Positive index values indicate favorable environments, while negative values indicate unfavorable ones [7]. Trials 12, 9, 4, 10, 8, and 6 were classified as unfavorable environments, while trials 5, 2, 3, 7, 1, 11, and 13 were favorable. In addition, we added a column with the standardized environmental index value.

We found that the different criteria (AIC, BIC, and PAL) pointed to different models as having a better fit. The AIC criterion identified model Leg.6.D, which has a diagonal structure for the residuals and an order six for the Legendre polynomials, as having the best fit (Table 2). The BIC and PAL criteria however, identified the Leg.5.D model as having the best fit. The AIC and BIC criteria prioritize, respectively, efficiency and consistency in their choices of

**Table 2. Different fitted models using the Legendre polynomials (Leg).**

| Model[1] | Order | P[2] | AIC | BIC | PAL | LRT |
|---|---|---|---|---|---|---|
| Leg.0.H | 0 | 2 | 11306.5 | 11318.9 | 11302.5 | 377.7* |
| Leg.1.H | 1 | 4 | 11287.6 | 11312.4 | 11279.6 | 400.6* |
| Leg.2.H | 2 | 7 | 11255.3 | 11298.8 | 11256.8 | 438.8* |
| Leg.3.H | 3 | 11 | 11218.1 | 11286.4 | 11229.1 | 484.1* |
| Leg.4.H | 4 | 16 | 11191.4 | 11290.9 | 11217.4 | 520.7* |
| Leg.5.H | 5 | 22 | 11149.6 | 11286.3 | 11197.3 | 574.5* |
| Leg.6.H | 6 | 29 | 11134.4 | 11314.6 | 11243.4 | 603.7* |
| Leg.0.D | 0 | 14 | 11006.7 | 11093.7 | 10978.7 | 507.6* |
| Leg.1.D | 1 | 16 | 10983.4 | 11082.8 | 10951.4 | 534.9* |
| Leg.2.D | 2 | 19 | 10928.4 | 11046.5 | 10930.0 | 595.9* |
| Leg.3.D | 3 | 23 | 10865.1 | 11008.1 | 10885.4 | 667.2* |
| Leg.4.D | 4 | 28 | 10739.6 | 10913.6 | 10778.8 | 802.7* |
| Leg.5.D | 5 | 34 | 10648.1 | 10859.4 | 10729.9 | 906.2* |
| Leg.6.D | 6 | 41 | 10645.4 | 10900.2 | 10880.5 | 922.9* |

[1]These models can assume homogeneous (H) or diagonal (D) residual variance structure. [2]Number of parameters

*Significant with the LRT test.

model [47, 48]. Corrales et al. [48], using simulated data, reported that when the true model was among the candidate models, the PAL and BIC criteria selected the same model. Furthermore, when the PAL and AIC criteria were used, the model selection was not always the same. When the real model was unknown, the AIC was more precise in choosing the best model, compared to the BIC. According to Vrieze [49], for very complex models (which include a high number of parameters) the BIC criterion was preferred over the AIC. Corrales et al. [48] stated that the PAL criterion simultaneously considers the consistency and efficiency of a model and should, therefore, be preferred over the AIC and BIC criteria when choosing models. The model ultimately chosen was Leg.5.D.

Based on the chosen model (Leg.5.D), the random effects of the cultivars were modeled as linear functions using the Legendre polynomials, with order five and heterogeneous residual variance (diagonal). This resulted in 34 estimated parameters, 13 of which were associated with residuals, that is, one for each trial, and 21 related to the model's genotypic components. It is of note that the genetic effect was significant with the LRT test for all fitted models, indicating high variability between the cultivars evaluated (Table 2).

The average accuracy for the prediction of the genotypic values for each cultivar, based on the Leg.5.D model, are shown in Fig 1. We found that the average accuracy of predictions was greater when more trials were used to evaluate the cultivars. The accuracy observed for the cultivars that were present in the 13 environments was the highest, while the accuracy estimates for the cultivars evaluated in only two environments were the lowest. The accuracy values of each cultivar in each environment are available in S4 Table (supporting information).

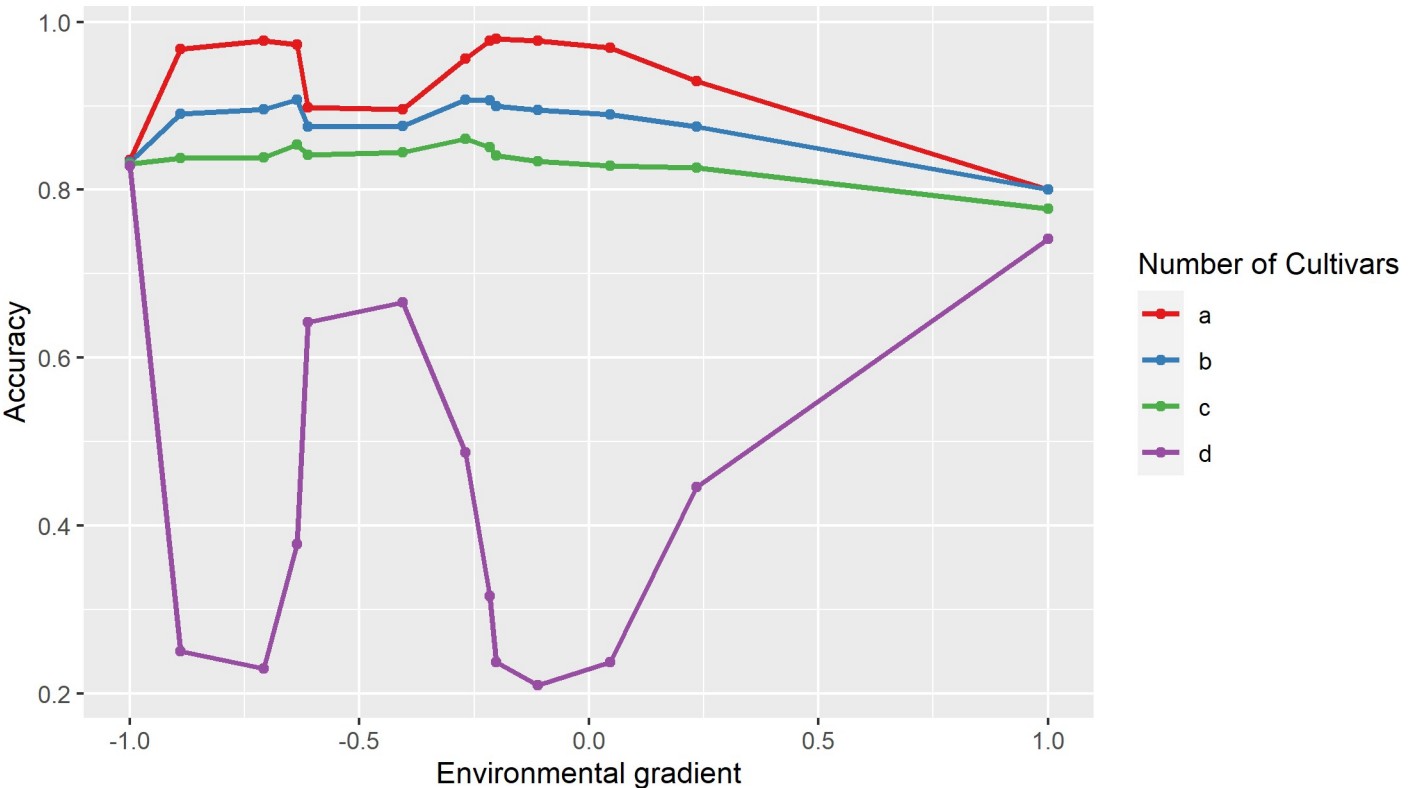

**Fig 1. Average accuracy of the prediction in each trial for the genotypic values of the cultivars.** a) Cultivars evaluated in 13 trials (80 cultivars); b) cultivars evaluated in nine trials (20 cultivars); c) cultivars evaluated in six trials (four cultivars); and d) cultivars evaluated in only two trials (one cultivar). The trials are ordered according to the standardized environmental index (Table 1).

Using the RRM, the adaptability and stability of 100 of the 105 cultivars was quantified as a reaction norm. These 100 cultivars were evaluated in at least nine of the 13 trials, with the accuracy in predicting their genotypic values, equal to or greater than 80%, including for those trials in which the cultivars were not evaluated (S4 Table).

According to Eq 10, the cultivars were recommended by comparing them with the three proposed ideotypes (three scenarios): cultivars of general adaptability, cultivars of maximum adaptability to unfavorable environments, and cultivars of maximum adaptability to favorable environments. The probability values of each cultivar in each scenario are presented in S5 Table.

Fig 2 shows the reaction norm curves of the ten common bean cultivars with the highest potential (highest probability value), considering the general adaptability scenario (ideotype—maximum performance genotypes in both unfavorable and favorable environments), as well as the cultivars used as checks (Pérola and Ouro Negro). The probability of each cultivar was calculated according to Eq 10, in relation to the ideotype for the scenario of general adaptability. Among the ten selected cultivars, six had the Carioca grain type (BRS Estilo, IAC Formoso, IAC Imperador, IPR Andorinha, IPR Campos Gerais and VC 15), and four had the Black grain type (BRS Agreste, IPR Tiziu, IPR Tuiuiú and VP 22). The IPR Campos Gerais cultivar surpassed the Pérola cultivar in all trials, while the VP 22 cultivar surpassed the Ouro Negro cultivar in all trials.

The reaction norm curves of the ten common bean cultivars with the greatest potential (highest probability value), considering the scenario of maximum adaptability to unfavorable environments (ideotype—maximum performance genotypes in unfavorable environments,

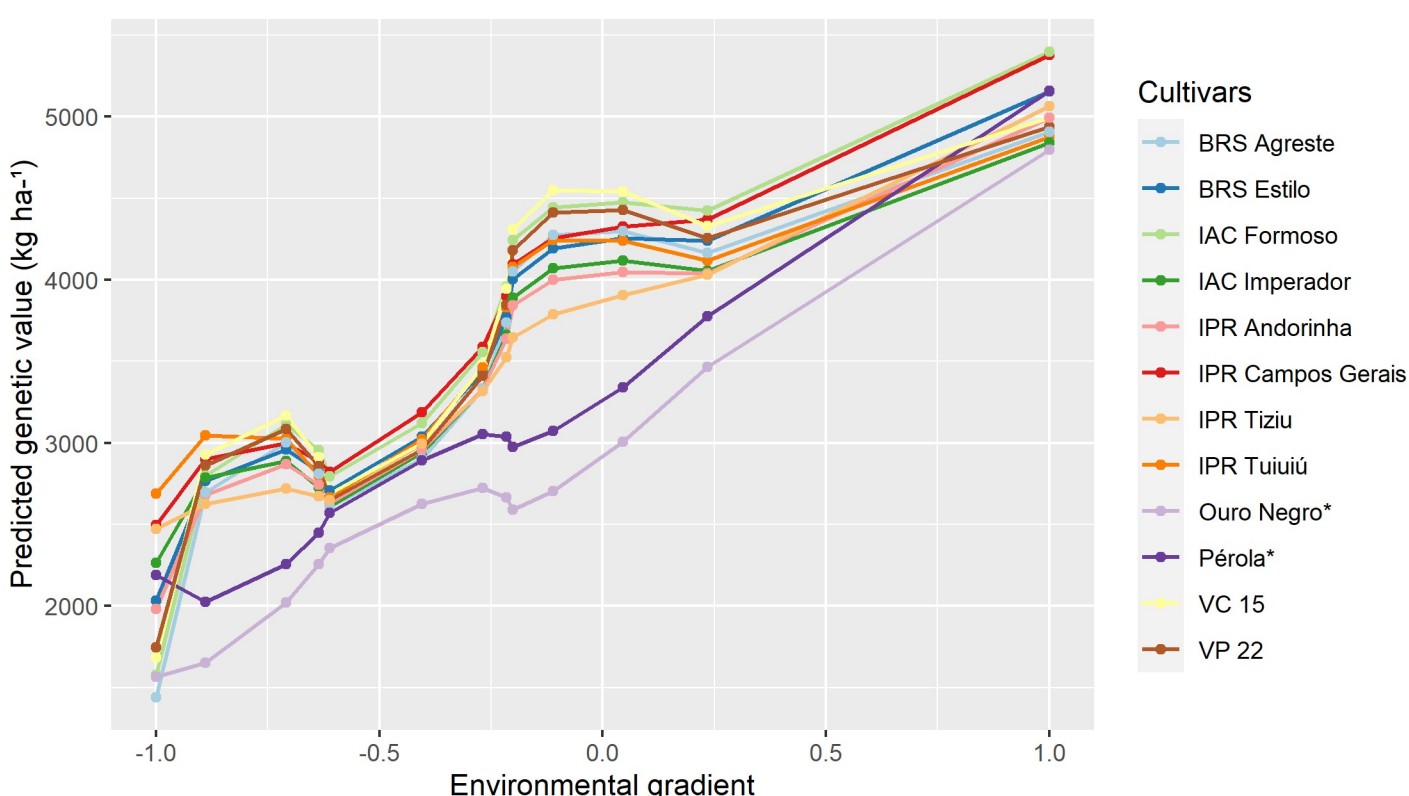

**Fig 2. Cultivars of Carioca and Black common bean of general adaptability according to the ideotype.** The trials are ordered according to the standardized environmental index (Table 1). *Cultivars used as checks.

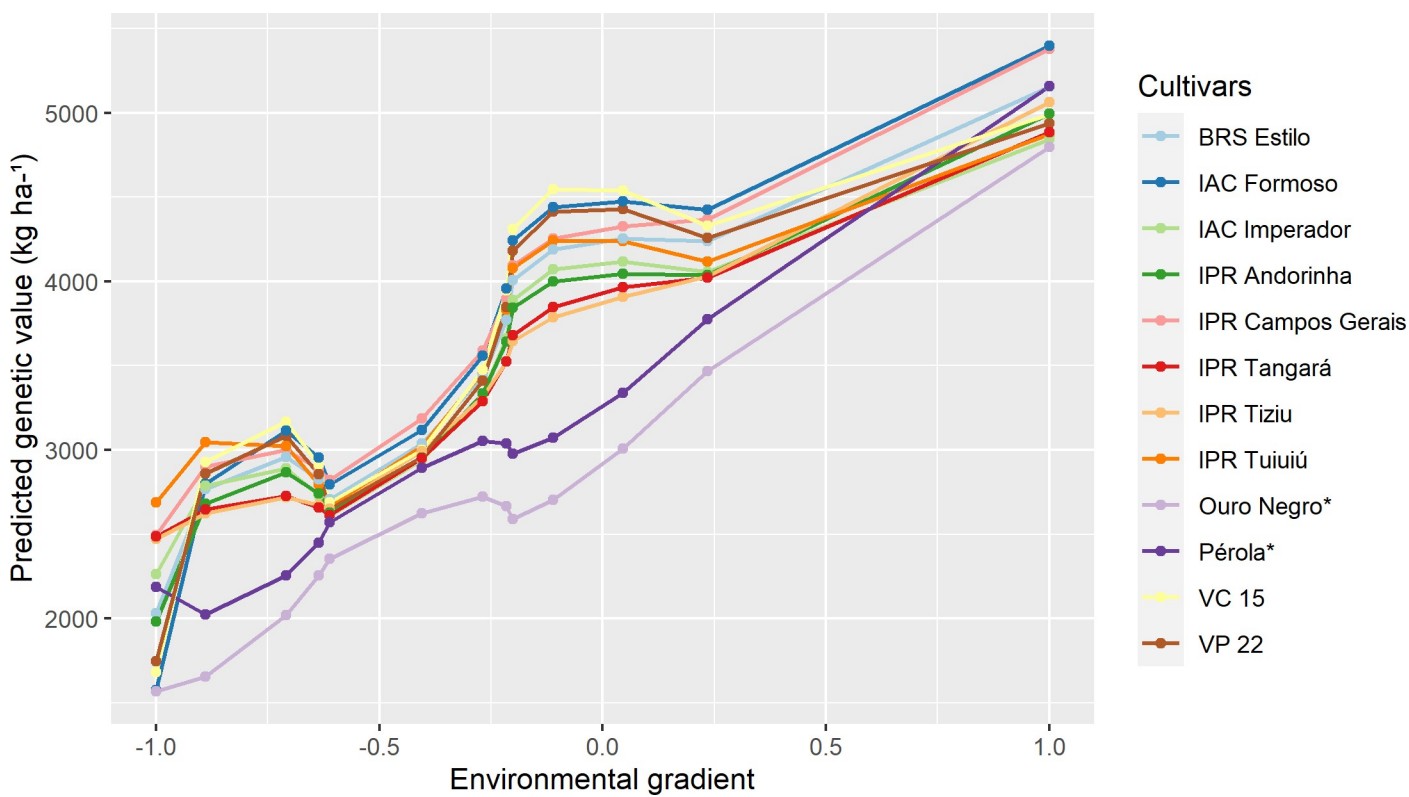

**Fig 3. Cultivars of Carioca and Black common bean of maximum adaptability for unfavorable environments according to the ideotype.** The trials are ordered according to the standardized environmental index (Table 1). *Cultivars used as checks.

regardless of their performance in favorable environments), as well as the cultivars used as checks, are presented in Fig 3. Of the ten selected cultivars, seven had Carioca grain (BRS Estilo, IAC Formoso, IAC Imperador, IPR Andorinha, IPR Campos Gerais, IPR Tangará and VC 15) and three had Black grain (IPR Tiziu, IPR Tuiuiú and VP 22). The cultivar IPR Campos Gerais surpassed the cultivar Pérola in all trials, and the IPR Tuiuiú, IPR Tiziu, and VP 22 cultivars exceeded the Ouro Negro cultivar.

In Fig 4, the reaction norm curves for the ten cultivars with the highest potential (highest probability value), considering the scenario of maximum adaptability to favorable environments (ideotype—maximum performance genotypes in favorable environments, regardless of their performance in unfavorable environments), as well as the cultivars used as checks, are shown. Of the ten selected cultivars, seven had Carioca grain (BRS Estilo, IAC Formoso, IAC Imperador, IPR Andorinha, IPR Campos Gerais, IPR 139 and VC 15) and three had Black grain (IPR Agreste, IPR Tuiuiú and VP 22). The IPR Campos Gerais cultivar surpassed the Pérola cultivar, in all trials, and the IPR Agreste, IPR Tuiuiú, and VP 22 Black common bean cultivars exceeded the Ouro Negro cultivar, in all trials.

## Discussion

Rating the variations of a set of trials, according to an environmental gradient, is essential when using methods based on linear regression that aim to quantify the adaptability of a cultivar. Finlay and Wilkinson [5] proposed using the average performances of the cultivars in each trial as a gradient, and estimating an environmental index using the differences between the average of the cultivars evaluated in each trial and the general average of the cultivars in all

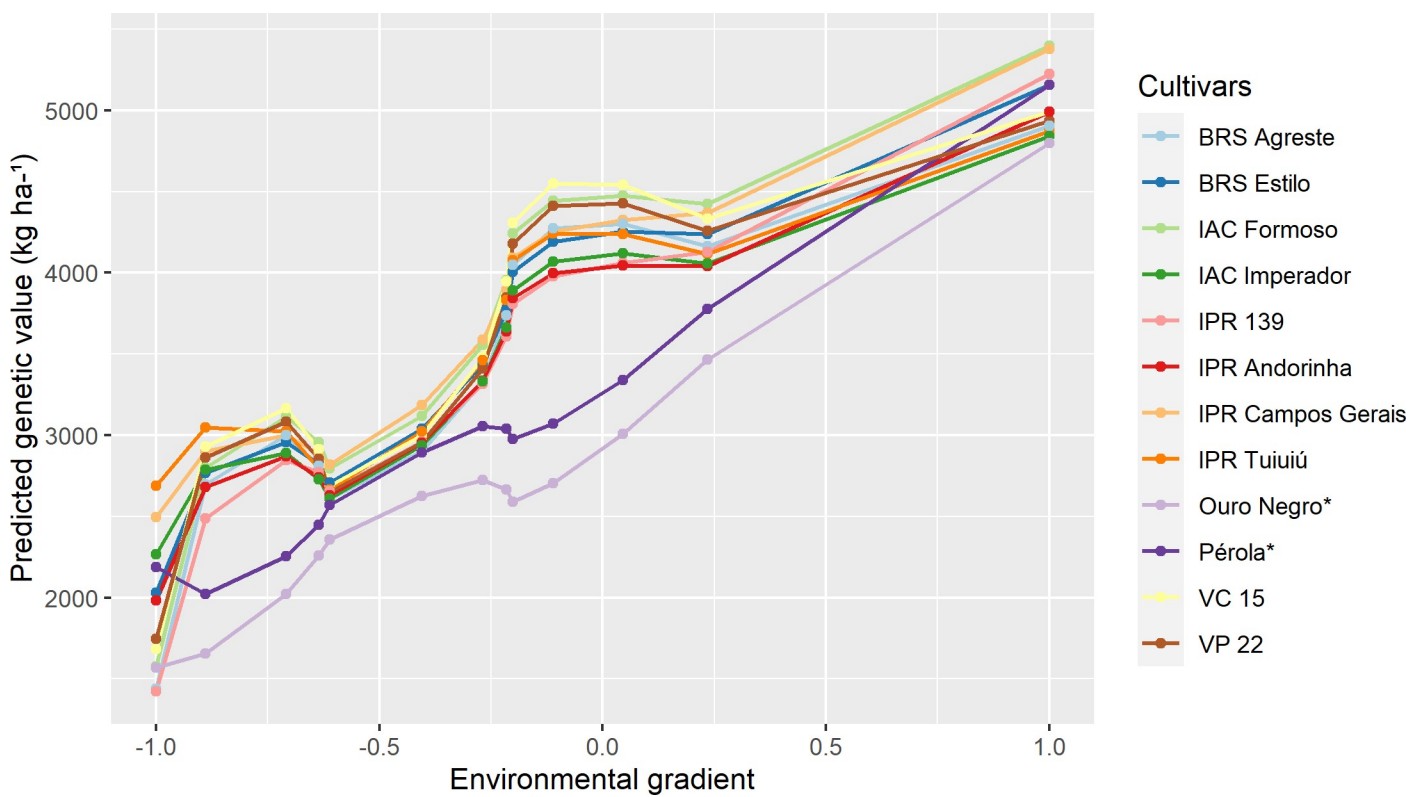

**Fig 4. Cultivars of Carioca and Black common bean of maximum adaptability for favorable environments according to the ideotype.** The trials are ordered according to the standardized environmental index (Table 1). *Cultivars used as checks.

trials. Additionally, the fit of the regression model for each cultivar was made according to its performance, relative to the environmental index, in order to increase the values. The lack of an environmental gradient complicates the interpretation of the behavior of the genotypes in the face of the environmental variations [5]. Thus, the ordering of environments along a gradient rather than a set of arbitrarily defined groups of the data is necessary in reaction norm models, because these models describe the phenotype of an individual expressed as a function of the gradual and continuous change in environments [50, 51].

When classifying the trials with the environmental index (in favorable or unfavorable environments), it was observed that the seasons, places, and years in which the trials were conducted did not determine the classification, as the trials from the same place and year could have very different results (trials 7 and 9), while those from different seasons, places, and years could be very similar (for example, environments 1 and 11). It should be noted that trial 9 was planted 44 days after trial 7, which may be one of the justifications for the different environmental index values. These results could be caused by edaphoclimatic variations, as well as variations in the incidence of pests and diseases in the environments in which the cultivars were evaluated, resulting in GEI. Several authors have also previously [52–55] reported the influence of these factors on the environmental classification, resulting in significant GEI. For Ramalho et al. [56], the most significant contributions to the GEI in the common bean culture were due to the combinations of cultivar × season and cultivar × years.

The development of methods to model GEI is coupled with the availability of more genotypic and environmental information, in line with the advances in data collection and analyses. The first analyses were based on analyses of variance [57, 58], with a single parameter to

interpret the adaptability and stability. The advances with the development of new methodologies however, are based on regression analyses, with interpretations based on more parameters, such as the average, the regression coefficient, the regression deviation, and new definitions of adaptability and stability [5, 8, 14].

Currently, the effects of genotypes and environmental conditions can be modeled by phenotypic values in regression with genetic markers and in environmental covariates, via mixed models [59]. However, these models consider that the genotype behavior is linear, which may not equate to the genotypes actual behavior. Thus, RRM in conjunction with Legendre polynomials are used to establish the order of the polynomials of the regression parameters later, according to the behavior of the genotypes in a MET. Additionally, the mixed model approach also allows for the genotypic values of individuals to be predicted, as adaptability and stability are genotypic, and not phenotypic.

According to Ni et al. [60], reaction norm models allow for the adjustment of an individual's genetic effects with their exposure to the environmental effects, so that the genotypes are adjusted as a nonlinear function of a continuous environmental gradient. The adjustment of reaction norm models, as a function of the environmental gradient, considering Legendre polynomials, captures more adequately the behavior of the genotypes in a MET. These fact is an advantage of the reaction norms in relation to the traditional methods of analyses of adaptability and stability.

The inclusion of kinship information between the individuals evaluated, whether via pedigree or genomics, could contribute to the use of RRM, which is based on the estimation of the variance components using the method of Restricted Maximum Likelihood associated with the Best Linear Unbiased Predictor (REML/BLUP) [61]. This information can be incorporated into the incidence matrix of the genetic values in the matrix model, allowing more accurate estimates, and consequently, increasing the prediction accuracy. Several studies are available that show that the inclusion of kinship information provides better adjusted models and lower values of residual variance estimates [27, 62–65].

In addition, the availability of environmental information, such as temperature, soil moisture, and rainfall data, can help in estimating the environmental values used in the reaction norms. Ly et al. [28] showed in their work that much of the variation caused by the GEI is owing to the changes caused by environmental covariates. One of the most standard ways of adding spatial variations to a statistical model is through structures of spatial variance and covariance, as proposed by Cullis and Gleeson [66] and later refined by Gilmour et al. [67]. Furthermore, several studies have incorporated the effects of environmental covariates in their statistical models, through the inclusion of random effects in the incidence matrices [27, 28, 68, 69].

Jarquín et al. [59] state that it is possible to simultaneously model the effects of environmental covariates and the genomic data obtained. However, this approach would lead to very demanding analyses with a high number of parameters. Thus, these authors proposed models of random effects where the effects of markers and environmental covariates are modeled together, through covariance structures, which can significantly improve the prediction accuracy.

To quantify the adaptability and stability using reaction norms, the prediction accuracy represents the reliability in the evaluation of the behavior of the evaluated genotypes in different environments. In this work, most of the accuracy estimates obtained for each cultivar in each environment were greater than 80%, which also resulted in an average accuracy of the 13 trials that was higher than this value. In the VCU trials, Resende and Duarte [43] recommended that the accuracy should be at least 80%. Other previous investigations have also highlight the importance of prediction accuracies, using the reaction norm models in plant breeding experiments [59, 70, 71].

Another advantage of using reaction norm is the prediction of genotypic values for the cultivars for environments in which they were not evaluated, when the MET presents genetic imbalance. When using trials with unbalanced data, or just a sample of the cultivars, the prediction accuracy estimates tend to be lower, and the model may not be efficient in evaluating the performance of the cultivars [72, 73]. Viana et al. [74] working with unbalanced data, showed that the lack of data resulted in relevant effects on the estimation of genetic variances, the accuracy of prediction and the ranking of predicted genetic values, which can affect the efficiency of selection.

For Smith et al. [1], using accurate information for the behavior of the cultivars, allowed breeders to choose the best varieties, according to the needs of farmers, in order to maximize profitability and food security. One of the difficulties in assessing the behavior of a group of cultivars over MET was due to the fact that new genotypes were included in the trials over the years, in addition to the loss of information due to problems that occurred over the trials, resulting in genetic and statistical unbalanced.

As noted, only 12 cultivars of superior performance were found in Figs 2–4, with eight Carioca bean cultivars and four Black bean cultivars, instead of 30 cultivars (10 per figure). This was because there were some cultivars that were widely adaptable and highly stable that were selected for more than one scenario, such as the IPR Campos Gerais and IPR Tuiuiú.

Cultivars with high phenotypic averages for high yield were identified, but they were not included in Figs 2–4, as those selected by the reaction norm models. This can be explained by the fact that the methodology when calculating the probability of each cultivar that was based on the cultivar-ideotype distance penalizes cultivars that showed great variation in their productivity during the trials, even if they presented high general averages. Thus, the reaction norm models can also quantify the stability of cultivars, defined as the variation regarding the behavior of the ideotype across environments. Eeuwijk et al. and Van Oijen and Höglind [11, 75] also reported this property of reaction norms. It is also worth mentioning that the use of the ideotype that was established from the data itself, had the advantage of comparing the genotypes with a real situation observed for that MET, since the ideotype is defined as the maximum value predicted in each trial.

The reaction norms, based on RRM, can also model the heterogeneity of the genetic variations and correlations between the environments, in addition to the spatial trends in the trials [16]. Furthermore, these models allow for more accurate estimations of the genotypes in the trials, as well as better estimations of the genetic parameters, such as heritability, variances, covariances, and genetic correlations, while they become more difficult in models with only fixed effects [11].

The maintenance of productivity in different environments is explained by the response to the environmental stimulus, being caused by the differential expression of the genes present in each individual. In this way, the adaptability and stability indicated in the reaction norm curves of the cultivars, provides information regarding their capacity to express phenotypes that may better adjust to the environmental conditions [76]. In this sense, one way to improve the adaptability of cultivars to the different environments in which they will be cultivated, is to pyramid the genes of maximum expression in both the unfavorable and favorable environments. The superior cultivars in each studied scenario were developed in different breeding programs from four institutions (EMBRAPA, UFV, IAC, and IAPAR). This is indicative of the effort and success of these breeding programs, as well as the genetic diversity between them, since the breeding programs are independent, with their own parental lines. Several studies have reported the decrease of genetic diversity in crops with genetic breeding, including common beans [77–80]. The hybridization between cultivars developed by different programs and adapted to different locations may result in the maintenance of genetic diversity and the

possibility of gains with the selection. Thus, these cultivars evaluated in this work also have the potential to be used in common bean breeding programs.

Finally, we can summarize that the use of reaction norm models associated with the Legendre polynomials, allows to adjust the behavior of the genotypes along a MET (as a function of a gradient). The methodology has the capacity to predict the genotypic values of individuals, even in places without phenotypic data, heterogeneity of genetic variations, correlations between environments, and spatial trends in the trials. Then, from the genotypic values and using an ideotype, it is possible to estimate the adaptability and stability of individuals.

## Conclusion

The random regression models to evaluate the adaptability and stability of cultivars appears to be an alternative in the evaluation of multi-environment trials, because it allows you to deal with unbalanced data, as well an improved evaluation of cultivar behavior.

The cultivars IPR Campos Gerais, IAC Formoso and VC 15 were the most adapted to the scenario of general adaptability, while the cultivars IPR Campos Gerais, IPR Tuiuiú and BRS Estilo performed better in unfavorable environments and the cultivars IAC Formoso, IPR Campos Gerais and VC 15 were better in places with favorable conditions.

## Supporting information

**S1 Table. Carioca bean cultivars, institutions of origin and year of recommendation.**
(DOCX)

**S2 Table. Black bean cultivars, institutions of origin and year of recommendation.**
(DOCX)

**S3 Table. Description of the trials.**
(DOCX)

**S4 Table. Accuracy of 105 cultivars in each trial.**
(DOCX)

**S5 Table. Recommendation probability values for each cultivar in each scenario.**
(DOCX)

**S1 Code. Script for analyses.**
(DOCX)

**S1 DOI. Dataset availability.**
(DOCX)

## Acknowledgments

We would like to thank the students of the Programa Feijão for their contribution with the help of data collection for this work. We would like to thank Editage (www.editage.com) for English language editing.

## Author Contributions

**Conceptualization:** Michel Henriques de Souza.

**Data curation:** Michel Henriques de Souza, José Domingos Pereira Júnior, Skarlet De Marco Steckling, Jussara Mencalha, Fabíola dos Santos Dias, José Eustáquio de Souza Carneiro.

**Formal analysis:** Michel Henriques de Souza, João Romero do Amaral Santos de Carvalho Rocha, Pedro Crescêncio Souza Carneiro.

**Funding acquisition:** Michel Henriques de Souza, José Eustáquio de Souza Carneiro.

**Investigation:** Michel Henriques de Souza, José Domingos Pereira Júnior, Skarlet De Marco Steckling, Jussara Mencalha, Fabíola dos Santos Dias, João Romero do Amaral Santos de Carvalho Rocha, Pedro Crescêncio Souza Carneiro, José Eustáquio de Souza Carneiro.

**Methodology:** Michel Henriques de Souza, João Romero do Amaral Santos de Carvalho Rocha, Pedro Crescêncio Souza Carneiro, José Eustáquio de Souza Carneiro.

**Project administration:** Michel Henriques de Souza, José Eustáquio de Souza Carneiro.

**Resources:** Michel Henriques de Souza, Pedro Crescêncio Souza Carneiro, José Eustáquio de Souza Carneiro.

**Software:** João Romero do Amaral Santos de Carvalho Rocha.

**Supervision:** Michel Henriques de Souza, Pedro Crescêncio Souza Carneiro, José Eustáquio de Souza Carneiro.

**Validation:** Michel Henriques de Souza, Skarlet De Marco Steckling, Pedro Crescêncio Souza Carneiro, José Eustáquio de Souza Carneiro.

**Visualization:** Michel Henriques de Souza.

**Writing – original draft:** Michel Henriques de Souza, João Romero do Amaral Santos de Carvalho Rocha.

**Writing – review & editing:** Michel Henriques de Souza, Pedro Crescêncio Souza Carneiro, José Eustáquio de Souza Carneiro.

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
