## [Decision Letter · Decision Letter 0]

8 Jun 2020

PONE-D-20-12530

New genotypic adaptability and stability analyses using Legendre polynomials and genotype-ideotype distances

PLOS ONE

Dear Dr. de Souza,

Thank you for submitting your manuscript to PLOS ONE. After careful consideration, we feel that it has merit but does not fully meet PLOS ONE’s publication criteria as it currently stands. Therefore, we invite you to submit a revised version of the manuscript that addresses the points raised during the review process.

We look forward to receiving your revised manuscript.

Kind regards,

Roberto Fritsche-Neto, Ph.D.

Academic Editor

PLOS ONE

Journal Requirements:

4. Please amend your manuscript to include your abstract after the title page.

Reviewers' comments:

Reviewer's Responses to Questions

**Comments to the Author**

1. Is the manuscript technically sound, and do the data support the conclusions?

Reviewer #1: Yes

Reviewer #2: Partly

2. Has the statistical analysis been performed appropriately and rigorously? 

Reviewer #1: Yes

Reviewer #2: No

3. Have the authors made all data underlying the findings in their manuscript fully available?

Reviewer #1: Yes

Reviewer #2: No

4. Is the manuscript presented in an intelligible fashion and written in standard English?

Reviewer #1: Yes

Reviewer #2: Yes

5. Review Comments to the Author

Reviewer #1: Overview

The proposed methodology could be an alternative to traditional models in the study of MET. It uses the phenotypic values of the genotypes across trials to create an environmental gradient and fits a reaction norm model using Legendre polynomials to predict the genotype performance. Although, using only the means of genotypes and general mean to create the environmental gradient may not be best alternative due disregard edaphoclimatic conditions.

The use of ideotypes to study the adaptability and stability seems to be a good alternative. It is possible to obtain four classes of ideotypes and from this select the desired genotypes.

A few considerations must be pointed:

The author may hypothesize why using Environmental gradient applying Finlay and Wilkinson is advantageous than environment traits to obtain the Environmental gradient;

The authors did not compare the proposed methodology to established methodologies. This could enhance the discussion;

Is it possible to use covariance matrix for the genotypic effect? It was used in the Phaseolus vulgaris L application?

A detailed description follows below:

Introduction:

Introduction underline the importance of genotype x environment studies. Although, it lacks reference that reinforce the proposed methodology and its theoretical background.

Line 35-36: Check the reference

Line 47-54: Need reference for theses affirmations

Line 66: imbalances or unbalanced? check in the text

Methods description (step by step):

Well written and understandable.

Application of the method with Phaseolus vulgaris L.

Results

Line 240: humidity to moisture

Line 267: witnesses to check

Discussion

Usually, in animals and plants, Legendre polynomials and other random regression are used to model the growth of the individual. In the presented methodology, it was not clear the “biological” meaning of modeling reaction norm using only environmental gradient with only the same priori information. see DOI 10.1007/s00122-013-2243-1. How the model deal with genotype x environment interaction? Is it feasible to extrapolate the presented methodology to unobserved environments? Since the Mixed model approach is used.

Line 424-425: How the presented methodology could incorporate the genomic matrix and environmental data in order to increase predictive ability?

Line 429-430: Which studies are these? What are the connections between the presented methodology and them?

Line 431-437: “The fact that an individuals' behavior is not predetermined, is an advantage of the proposed methodology in relation to the traditional methods of analysis of adaptability and stability.” I do not understand this affirmation. The model doesn’t use the environmental gradient defined a prioi from observed genotypes?

Line 446-454: Advantage over which other methodology? The presented work did not compare with the most common methodologies applied.

Line 452-454: It was not clear if it was used a cross-validation scenario to obtain these predictive accuracies. If it wasn’t, interpretation regarding predictive accuracy must be pondered.

Line 460-461: Comparison among mixed models and others approach must be careful since the present work did not evaluate any other approach.

Line 482-524: These paragraphs focus mainly on bean recommendation. Although very important, these do not seem to be the main focus of the study.

Reference

Many references are in Portuguese or are from books. These type of refences should be avoided in order to ensure access to the readers.

Reviewer #2: The authors applied random regression models (RRM) to investigate the reaction norms in common bean cultivars. This is a timely topic because understanding the genotype by environment is increasingly becoming important given the recent climate change. Although the topic discussed here is fascinating, there are a number of subjects that the authors may want to expand.

The authors argued as if they developed RRM to investigate phenotypic plasticity  or reaction norm. For instance, this is reflected in the title, "We propose a new methodology" in the Abstract, and "Thus, the objectives of this investigation were to propose a new methodology for ..." However, this is incorrect as the use of RRM for phenotypic plasticity analysis has been an active research area in animal breeding and ecology for more than decades. You are applying the previously developed methodology to study phenotypic plasticity in plants by treating multi-environment trials as environmental gradients. This manuscript would be much more appealing to plant breeders if written in the context of those works by others.

My second reservation pertains to the Introduction section that lacks a good structure. First, start by framing why adaptability/stability analysis is an important research area in the context of multi-environment trials (MET) and genotype by environment (GEI). Note that MET and GEI were never mentioned in the Introduction in the current version. The term MET first appears in the Methods section and GEI is only referred to in the Discussion section. Second, briefly review rich literature on how and why RRM has been widely used to study plasticity or GEI in animal breeding and ecology. Lastly, clearly state the objective of this paper, which is the application of RRM to investigate reaction norms in plant breeding. Also, just saying linear mixed models coupled with Legendre polynomials is not sufficient. Explicitly mention that the model you are using is RRM. I suggest entirely rewriting the Introduction.

The third major concern is the lack of methodological clarity and smoothness. The current Methods section is written like a step by step software manual with a disconnection between them. Not quite sure how Ij defined in step 1 is relevant to the subsequent steps. Steps in L131 and L144 can be combined with others because they are well-known statistics. I believe the methodology part can be significantly improved.

The utility of RRM is limited in this study because the authors did not use any genetic data (pedigree or genomics). Consider adding one or two paragraphs, regarding the impact of genetics on adaptability and stability analyses in the Discussion section.

Below are my specific comments.

Data availability statement. It is a bit strange to say that the data will be available upon acceptance of this manuscript. The data should be made available for reviewers and editors through the reviewing process if you intend to make them open.

L47: It was not clear to me what you are referring to by current technologies and lower level technologies.

L123: How is the vector of phenotypes constructed? Is it ordered by genotypes or trials? This will determine how you specify the variance-covariance terms for g and e. There is an inconsistency in your notation because Kg is on the left side of the Kronecker product, while Sigma_e is on the right side of the Kronecker product.

L155, L163, L168: It is misleading to say "at the original scale". You are just predicting breeding values from estimated random regression coefficients.

L157: What BLUPs are you referring to? I do not see any connection between this and the preceding subsections.

L173: Clarify how PEV was computed from a random regression model.

Equation 9: A reference is needed.

L204: I did not understand this paragraph. What does it mean by "through the invariance in multi environment trials (MET)."?

Genetic material (L216) and Trials (L223) should be placed before the Methods description (L83) according to the flow.

L218: How population structure of Carioca grains and Black grains were accounted for. I believe this will impact the interpretation of results.

L322: It is not evident in Figure 1, which ones are environmental gradients six and eight.

L324: This is essential information. I suggest you show a table or a figure related to variance components.

Figure 1: Clarify why the x-axis is ranging from -1 to 1. This does not agree with Table 1.

L340: You previously mentioned in L264 that the cultivars of minimal adaptability will not be considered for recommendations. Why do the cultivars of minimal adaptability appear here again?

L406: Introduce the concept of genotype by environment interactions first in the Introduction section rather than in the Discussion.

L426: This paragraph should be placed in the Introduction section.

L446: You did not propose the method (reaction norm analysis using RRM), but you applied.

Minor comments.

L117: Consider replacing order of adjustment with order of polynomials

L122, L293, L308: residue -> residual or residuals

L200: Why na appears twice for k = 1 and k = 4?

L281: table 1 -> Table 1.

L293: Consider replacing grade six with order six. Also, see L306.

Table 2: The column LOG L is redundant if all models converged.

L313: There is no need to redefine DEG, AIC, BIC, PAL, LRT, H, and D again.

L322: figure 1 -> Figure 1

L401-402: Be consistent with the use of trials vs. environments

6. PLOS authors have the option to publish the peer review history of their article (what does this mean?). If published, this will include your full peer review and any attached files.

Reviewer #1: No

Reviewer #2: No

---

## [Author Response · Author response to Decision Letter 0]

17 Jul 2020

The authors of this paper would like to thank the reviewers' comments and suggestions. His excellent tips contributed greatly to the improvement of this work, leading to greater understanding and ease of interpretation. We carefully analyze each comment and seek to meet the requests of both reviewers. In the file named "response to reviewers", there is a detailed description of each suggestion and the change made to the article, in order to simultaneously serve the reviewers.

---

## [Decision Letter · Decision Letter 1]

27 Aug 2020

PONE-D-20-12530R1

Adaptability and stability analyses of plants using random regression models

PLOS ONE

Dear Dr. de Souza,

Thank you for submitting your manuscript to PLOS ONE. After careful consideration, we feel that it has merit but does not fully meet PLOS ONE’s publication criteria as it currently stands. Therefore, we invite you to submit a revised version of the manuscript that addresses the points raised during the review process.

We look forward to receiving your revised manuscript.

Kind regards,

Roberto Fritsche-Neto, Ph.D.

Academic Editor

PLOS ONE

Reviewers' comments:

Reviewer's Responses to Questions

**Comments to the Author**

1. If the authors have adequately addressed your comments raised in a previous round of review and you feel that this manuscript is now acceptable for publication, you may indicate that here to bypass the “Comments to the Author” section, enter your conflict of interest statement in the “Confidential to Editor” section, and submit your "Accept" recommendation.

Reviewer #1: All comments have been addressed

Reviewer #2: (No Response)

2. Is the manuscript technically sound, and do the data support the conclusions?

Reviewer #1: Yes

Reviewer #2: Yes

3. Has the statistical analysis been performed appropriately and rigorously? 

Reviewer #1: Yes

Reviewer #2: Yes

4. Have the authors made all data underlying the findings in their manuscript fully available?

Reviewer #1: Yes

Reviewer #2: Yes

5. Is the manuscript presented in an intelligible fashion and written in standard English?

Reviewer #1: Yes

Reviewer #2: Yes

6. Review Comments to the Author

Reviewer #1: I would like to thank the authors for having accepted my suggestions and or clarified my doubts. The application of random regressions models and ideotypes proved to be suitable in the studies of genotype by environment interaction (GEI) in beans. Although the authors did not use any genomic/pedigree data or environmental information, the methodologies used provide opportunities for further research regarding the use of environmental and genomics data using Random Regression models in multi environmental trials studies.

A detailed description with minor review follows below:

L.48: Replace “imbalance” to “unbalanced”

L.155: Replace “Genetic imbalance” to “Genetic Unbalanced”

L. 293: Replace "Witnesses" to "check"

L. 294: Replace "as witnesses to check" "as check"

L.248: Equation 9. Reference needed

L.381: Replace "Witnesses" to "check"

L. 657: Remove

Reviewer #2: The authors have addressed most of my comments adequately. Below are my additional comments to improve the readability and strengthen the manuscript.

L103: This is incorrect. Henderson proposed BLUP, linear mixed model, and mixed model equations, but not RRM. The initial form of RRM was first proposed by Kirkpatrick et al. 1990 [1] and later extended by Schaeffer and Dekkers 1994 [2] and Meyer and Hill 1997 [3].

[1] Kirkpatrick et al. 1990 Analysis of the inheritance, selection and evolution of growth trajectories. Genetics 124: 979–993.

[2] Schaeffer LR, Dekkers JCM. Random regressions in animal models for test-day production in dairy cattle. Proc 5th World Congress on Genetics Applied to Livestock Production; 1994; Guelph, 18:443-446.

[3] Meyer K, Hill WG. Estimation of genetic and phenotypic covariance functions for longitudinal or repeated records by restricted maximum likelihood. Livest Prod Sci. 1997; 47:185–200.

It is important to note that the current work is not the first time to apply RRM to univariate or multivariate plant breeding data. The authors failed to refer to the earlier work of Sun et al. 2017 [4], Ly et al. 2018 [5], Momen et al. 2019 [6], and Baba et al. 2020 [7] in the Introduction section. Put them after animal breeding literature you referred and mention their contribution/relevance with respect to the current work.

[4] Sun et al. Multitrait, random regression, or simple repeatability model in high-throughput phenotyping data improve genomic prediction for wheat grain yield. Plant Genome. 2017; 10. pmid:28724067

[5] Ly et al. Whole-genome prediction of reaction norms to environmental stress in bread wheat (Triticum aestivum 736 L.) by genomic random regression. F Crop Res. 2018;216.

[6] Momen et al. Predicting longitudinal traits derived from high-throughput phenomics in contrasting environments using genomic Legendre polynomials and B-splines. G3: Genes, Genomes, Genetics. 2019.  9:3369-3380.

[7] Baba et al. Multi-trait random regression models increase genomic prediction accuracy for a temporal physiological trait derived from high-throughput phenotyping. PLoS ONE 2020. 15(2): e0228118.

In sum, the introduction section should be framed in the context of earlier work by others.

L185: I believe you meant the lowest and highest averages rather than lower and higher averages.

L191-194: What do you mean by "adjusted"? Perhaps "adjusted" is not the best term to use here.  

L196, Equation 3: Why don't you include fixed random regression coefficients designed to capture the mean trajectory of environmental gradients? Almost all of the previous RRM literature include this term to account for the mean trend. 

L248: Provide a reference for equation 9.

L264: Remove "thus"

L293, L375: What witnesses are? I am not familiar with this term. I believe you meant checks?

Table 2: Clarify what H or D means (first column) in the table caption.

L765: Smith et al. 2015 has been cited twice. Ref 1 and Ref 73.

Ref 20. Wrong journal title. Should be Journal of Dairy Science.

Ref 38: It says "Available: file:///C:/Users/Sandrinho/Downloads/artículo_redalyc_253021631009.pdf". Note that this is the author's locale file on a computer. Readers will not have access to it.

Lastly, I agree with reviewer 2 that references are used to support or defend your claims you made in your research. Note that PLOS ONE has a wide range of readers from around the world. I could not assess many references such as 9, 38-39, 42, 48-52, 61, 72, 76 because they are not written in English. Consider replacing them, if possible, with more suitable references that everyone can read, understand, and discuss.

7. PLOS authors have the option to publish the peer review history of their article (what does this mean?). If published, this will include your full peer review and any attached files.

Reviewer #1: No

Reviewer #2: No

---

## [Author Response · Author response to Decision Letter 1]

1 Oct 2020

Response to Reviewers

Dear reviewers,

The authors of this paper would like to thank the reviewers' comments and suggestions. We believe that this tips contributed greatly to the improvement of this work, leading to greater understanding and ease of interpretation. We carefully analyze each comment and seek to meet the requests of both reviewers. Below is a detailed description of each suggestion and the change made to the article, in order to simultaneously serve the reviewers.

Reviewer #1)

• L.48: Replace “imbalance” to “unbalanced”

• L.155: Replace “Genetic imbalance” to “Genetic Unbalanced”

• L. 293: Replace "Witnesses" to "check"

• L. 294: Replace "as witnesses to check" "as check"

• L.248: Equation 9. Reference needed

• L.381: Replace "Witnesses" to "check"

• L. 657: Remove

Authors reply: We would like to thank you the reviewer for the suggestions. We think they made this work better. We have been made the changes in the text. 

Reviewer #2: 

• L103: This is incorrect. Henderson proposed BLUP, linear mixed model, and mixed model equations, but not RRM. The initial form of RRM was first proposed by Kirkpatrick et al. 1990 [1] and later extended by Schaeffer and Dekkers 1994 [2] and Meyer and Hill 1997 [3]. 

[1] Kirkpatrick et al. 1990 Analysis of the inheritance, selection and evolution of growth trajectories. Genetics 124: 979–993.

[2] Schaeffer LR, Dekkers JCM. Random regressions in animal models for test-day production in dairy cattle. Proc 5th World Congress on Genetics Applied to Livestock Production; 1994; Guelph, 18:443-446.

[3] Meyer K, Hill WG. Estimation of genetic and phenotypic covariance functions for longitudinal or repeated records by restricted maximum likelihood. Livest Prod Sci. 1997; 47:185–200.

Authors reply: We apologize for the mistake. In this way, we understood the suggestions pointed out by the reviewer and made changes to the text, as suggested. We would like to thank you for the suggestions.

• It is important to note that the current work is not the first time to apply RRM to univariate or multivariate plant breeding data. The authors failed to refer to the earlier work of Sun et al. 2017 [4], Ly et al. 2018 [5], Momen et al. 2019 [6], and Baba et al. 2020 [7] in the Introduction section. Put them after animal breeding literature you referred and mention their contribution/relevance with respect to the current work.

[4] Sun et al. Multitrait, random regression, or simple repeatability model in high-throughput phenotyping data improve genomic prediction for wheat grain yield. Plant Genome. 2017; 10. pmid:28724067

[5] Ly et al. Whole-genome prediction of reaction norms to environmental stress in bread wheat (Triticum aestivum 736 L.) by genomic random regression. F Crop Res. 2018;216.

[6] Momen et al. Predicting longitudinal traits derived from high-throughput phenomics in contrasting environments using genomic Legendre polynomials and B-splines. G3: Genes, Genomes, Genetics. 2019. 9:3369-3380.

[7] Baba et al. Multi-trait random regression models increase genomic prediction accuracy for a temporal physiological trait derived from high-throughput phenotyping. PLoS ONE 2020. 15(2): e0228118.

In sum, the introduction section should be framed in the context of earlier work by others.

Authors reply: Again, we made a mistake in not mentioning some works that used RRM previously. Then, we read these articles recommended by the reviewer and made the corrections in the text. Thank you for the suggestion.

• L185: I believe you meant the lowest and highest averages rather than lower and higher averages.

Authors reply: Yes, we wanted to mean the lowest and highest averages. Thank you for the suggestion. We changed these words in the text.

• L191-194: What do you mean by "adjusted"? Perhaps "adjusted" is not the best term to use here. 

Authors reply: It is true; the term adjusted was not the best in this sentence. We changed the word in the text to ‘fitted’.

• L196, Equation 3: Why don't you include fixed random regression coefficients designed to capture the mean trajectory of environmental gradients? Almost all of the previous RRM literature include this term to account for the mean trend.

• Thank for you doubt, reviewer. The reason why we did not use the coefficients for the fixed part of the model was that, specifically in this work, we were not interested in estimating a mean for a not existing environment throughout the MET. However, as the reviewer said that this is common in the literature, we also affirm that this could have been done, without problems for the model. It was just our option to choose this model, with the fixed part without the drawing of a trajectory. 

• L248: Provide a reference for equation 9.

Authors reply: We added the reference in the text for this equation.

• L264: Remove "thus"

Authors reply: the word “thus” was removed from the text.

• L293, L375: What witnesses are? I am not familiar with this term. I believe you meant checks?

• Authors reply: we accepted this suggestion and changed the word in the text to ‘checks’.

• Table 2: Clarify what H or D means (first column) in the table caption.

Authors reply: We put the D and H means in the table 2 caption. 

• L765: Smith et al. 2015 has been cited twice. Ref 1 and Ref 73.

Authors reply: We fixed those citations in the text. 

• Ref 20. Wrong journal title. Should be Journal of Dairy Science.

Authors reply: We fixed this reference in the text. 

• Ref 38: It says "Available: file:///C:/Users/Sandrinho/Downloads/artículo_redalyc_253021631009.pdf". Note that this is the author's locale file on a computer. Readers will not have access to it.

• Authors reply: We fixed this reference in the text. 

• Lastly, I agree with reviewer 2 that references are used to support or defend your claims you made in your research. Note that PLOS ONE has a wide range of readers from around the world. I could not assess many references such as 9, 38-39, 42, 48-52, 61, 72, 76 because they are not written in English. Consider replacing them, if possible, with more suitable references that everyone can read, understand, and discuss.

• Authors reply: thank you for the comment. We tried to change these references written in Portuguese and we did this for most of them. We just could not find works corresponding to references 38, 39 and 52, since they are very specific references to VCU trials and common beans trials in Brazil, with no similar articles in international journals.

These references are in the following order in the text:

[41] Resende MDV de, Duarte JB. Precisão e controle de qualidade em experimentos de avaliação de cultivares. Pesqui Agropecuária Trop. 2007;37: 182–194.

[42] Melo CLP de, Carneiro, José Eustáquio de Souza Carneiro PCS, Cruz CD, Barros, Everaldo Gonçalves de Moreira MA. Linhagens de feijão do cruzamento “Ouro Negro” x “Pérola” com características agronômicas favoráveis. Pesqui Agropecuária Bras. 2006;41: 1593–1598.

[54] Ramalho MAP, Abreu A de FB, Santos PSJ dos. Interações genótipos x épocas de semeadura, anos e locais na avaliação de cultivares de feijão nas regiões Sul e Alto Paranaíba em Minas Gerais. Ciência e Agrotecnologia. 1998;22: 175–181.

---

## [Decision Letter · Decision Letter 2]

20 Oct 2020

PONE-D-20-12530R2

Adaptability and stability analyses of plants using random regression models

PLOS ONE

Dear Dr. de Souza,

Thank you for submitting your manuscript to PLOS ONE. After careful consideration, we feel that it has merit but does not fully meet PLOS ONE’s publication criteria as it currently stands. Therefore, we invite you to submit a revised version of the manuscript that addresses the points raised during the review process.

We look forward to receiving your revised manuscript.

Kind regards,

Paulo Eduardo Teodoro, Dr.

Academic Editor

PLOS ONE

Additional Editor Comments (if provided):

Dear authors, your manuscript was returned to the initial reviewers. One accepted and another requested Minor Revision. I am also requesting some corrections. Answer the comments point-to-point so that I can make the Final Decision in the next round.

- Throughout the text use "common bean" instead of just "bean";

- quote all R packages used for the analyzes;

- In the Discussion, insert a final paragraph about what advances the proposed method has in relation to the dozens of methods in the literature;

- Add in the conclusions the best genotypes identified by RR models.

Reviewers' comments:

Reviewer's Responses to Questions

**Comments to the Author**

1. If the authors have adequately addressed your comments raised in a previous round of review and you feel that this manuscript is now acceptable for publication, you may indicate that here to bypass the “Comments to the Author” section, enter your conflict of interest statement in the “Confidential to Editor” section, and submit your "Accept" recommendation.

Reviewer #1: (No Response)

Reviewer #2: All comments have been addressed

2. Is the manuscript technically sound, and do the data support the conclusions?

Reviewer #1: Yes

Reviewer #2: (No Response)

3. Has the statistical analysis been performed appropriately and rigorously? 

Reviewer #1: Yes

Reviewer #2: (No Response)

4. Have the authors made all data underlying the findings in their manuscript fully available?

Reviewer #1: Yes

Reviewer #2: (No Response)

5. Is the manuscript presented in an intelligible fashion and written in standard English?

Reviewer #1: Yes

Reviewer #2: (No Response)

6. Review Comments to the Author

Reviewer #1: The authors addressed most of my comments. Follows some additional suggestions to improve the manuscript:

Please, check if the term "imbalances" is correct

I did not understand equation 9 and I did not find a direct connection between the citation. Considering that part of your conclusions is based on it, I recommend that you explain it better or reference it.

Reviewer #2: (No Response)

7. PLOS authors have the option to publish the peer review history of their article (what does this mean?). If published, this will include your full peer review and any attached files.

Reviewer #1: No

Reviewer #2: No

---

## [Author Response · Author response to Decision Letter 2]

13 Nov 2020

Dear editor and reviewers,

The authors of this paper would like to thank the reviewers' comments and suggestions. We believe that this tips contributed greatly to the improvement of this work, leading to greater understanding and ease of interpretation. We carefully analyze each comment and seek to meet the requests of both reviewers. Below is a detailed description of each suggestion and the change made to the article, in order to simultaneously serve the editor and the reviewers.

Editor:

- Throughout the text use "common bean" instead of just "bean";

Authors reply: Thank you for the suggestion. We made the changes in the text.

- quote all R packages used for the analyzes;

Authors reply: Thank you for the suggestion. We added the packages on the article and on the supporting information.

- In the Discussion, insert a final paragraph about what advances the proposed method has in relation to the dozens of methods in the literature;

Authors reply: Thanks for the suggestion. We added a paragraph at the end of the Discussion showing the advantages of the proposed method. In addition, over the Introduction and Discussion sections, we show a more detailed description of the method's advantages over those already available in the literature.

- Add in the conclusions the best genotypes identified by RR models.

Authors reply: Thank you for the suggestion. We did that.

Reviewer #1: 

Please, check if the term "imbalances" is correct

Authors reply: Thank you for the suggestion. We check that in the text and made some changes aiming a better understanding.

I did not understand equation 9 and I did not find a direct connection between the citation. Considering that part of your conclusions is based on it, I recommend that you explain it better or reference it.

Authors reply: Dear reviewer, thanks for the suggestion. We try to understand your doubts about the accuracy equation. We used the equation adapted from Gilmour et al. (1990), using the PEV (Predictor Error Variance) and Kg (covariance coefficients for genotypic effect) matrices. However, using Legendre orthogonal polynomials, we made an adaptation in the equation, pre and post multiplying the PEV and Kg by the Фijm matrix (Legendre's m-th polynomial for the j-th trial and the i-th genotype), in order to transform the values obtained from the Legendre scale to the original scale, as proposed by Kirkpatrick et al. 1990. In addiction, we add the two references cited below in the article.

[1] Gilmour, A.R., Gogel, B.J., Cullis, B.R., and Thompson, R. ASReml User Guide Release 3.0 VSN International Ltd, Hemel Hempstead, HP1 1ES, UK; 2009. www.vsni.co.uk

[2] Kirkpatrick M, Lofsvold D, Bulmer M. Analysis of the inheritance, selection and evolution of growth trajectories. Genetics. 1990;124: 979–993.

---

## [Editor Report · Decision Letter 3]

16 Nov 2020

Adaptability and stability analyses of plants using random regression models

PONE-D-20-12530R3

Dear Dr. de Souza,

We’re pleased to inform you that your manuscript has been judged scientifically suitable for publication and will be formally accepted for publication once it meets all outstanding technical requirements.

Kind regards,

Paulo Eduardo Teodoro, Dr.

Academic Editor

PLOS ONE

---

## [Editor Report · Acceptance letter]

19 Nov 2020

PONE-D-20-12530R3 

Adaptability and stability analyses of plants using random regression models 

Dear Dr. de Souza:

I'm pleased to inform you that your manuscript has been deemed suitable for publication in PLOS ONE. Congratulations! Your manuscript is now with our production department. 

Kind regards, 

on behalf of

Professor Paulo Eduardo Teodoro 

Academic Editor

PLOS ONE